# Irreversible Electroporation Mediates Glioma Apoptosis via Upregulation of *AP-1* and *Bim*: Transcriptome Evidence

**DOI:** 10.3390/brainsci12111465

**Published:** 2022-10-29

**Authors:** Shuangquan Yu, Lingchao Chen, Kun Song, Ting Shu, Zheng Fang, Lujia Ding, Jilong Liu, Lei Jiang, Guanqing Zhang, Bing Zhang, Zhiyong Qin

**Affiliations:** 1Department of Neurosurgery, Huashan Hospital Shanghai Medical College, Fudan University, Shanghai 200040, China; 2Intelligent Energy-Based Tumor Ablation Laboratory, School of Mechatronic Engineering and Automation, Shanghai University, Shanghai 200444, China; 3Division of Biomedical Engineering, University of Saskatchewan, Saskatoon, SK S7N 5A9, Canada; 4Biomedical Science and Technology Research Center, School of Mechatronic Engineering and Automation, Shanghai University, Shanghai 200444, China

**Keywords:** irreversible electroporation, glioma, apoptosis, regulatory cell death, *AP-1*, *FOS*, *JUN*, *Bim*

## Abstract

The heat-sink effect and thermal damage of conventional thermal ablative technologies can be minimized by irreversible electroporation (IRE), which results in clear ablative boundaries and conservation of blood vessels, facilitating maximal safe surgical resection for glioblastoma. Although much comparative data about the death forms in IRE have been published, the comprehensive genetic regulatory mechanism for apoptosis, among other forms of regulatory cell death (RCD), remains elusive. We investigated the electric field intensity threshold for apoptosis/necrosis (YO-PRO-1/PI co-staining) of the U251 human malignant glioma cell line with stepwise increased uniform field intensity. Time course samples (0–6 h) of apoptosis induction and sham treatment were collected for transcriptome sequencing. Sequencing showed that transcription factor *AP-1* and its target gene *Bim* (Bcl2l11), related to the signaling pathway, played a major role in the apoptosis of glioma after IRE. The sequencing results were confirmed by qPCR and Western blot. We also found that the transcription changes also implicated three other forms of RCD: autophagy, necroptosis, and immunogenic cell death (ICD), in addition to apoptosis. These together imply that IRE possibly mediates apoptosis by the *AP-1-Bim* pathway, causes mixed RCD simultaneously, and has the potential to aid in the generation of a systemic antitumor immune response.

## 1. Introduction

Glioblastoma multiform (GBM) accounts for the majority of gliomas (56%) [1] and displays the most aggressiveness. The efficacy of current surgery plus conventional radiation and chemotherapy for GBM is poor, despite a second surgical treatment [2,3]. GBM patients only have a median overall survival of 12–15 months [4] and inevitably die of tumor recurrence or progression. The development of novel treatment strategies is urgently required. As a novel non-thermal tumor ablation modality, irreversible electroporation (IRE) has successfully been applied to solid tumors, including liver cancer [5], pancreatic cancer [6], prostate cancer [7], and renal cell carcinoma [8], after Food and Drug Administration (FDA) approval in 2008. GBM is characterized by a rich blood supply and requires spatially maximal safe surgical resection with minimal disruption of neurological function. Compared to conventional thermal ablative technologies, IRE ablation would not be subject to heat-sink effects [9] from nearby blood flow and have clear ablation boundaries. Moreover, IRE effectively protects vascular networks within treated areas. These make it a suitable therapeutic candidate for GBM [10]. Within the field of GBM treatment, there was only modest evidence of animal research and in vitro cell experiments for IRE [11,12]. Further investigation into gene regulatory mechanisms and therapeutic efficacy after IRE must be completed.

Selectively based on specific signaling pathways, there have been numerous studies for gene regulatory mechanism of regulatory cell death (RCD), including apoptosis after IRE. Loss of mitochondrial membrane potential, cytochrome c release, caspase-9 activation, upregulation of pro-apoptotic factors (BAX, BAK, BAD), and downregulation of anti-apoptotic factors (Bcl-2, Bcl-xL, Mcl-1) confirmed the involvement of mitochondria, mostly through the intrinsic apoptosis pathway. However, in some studies, BID cleavage also points to the activation of type II extrinsic-like apoptosis. In some cells (HCT, B16F10, E4 SCC, Jurkat), apoptosis also progresses through the type I extrinsic-like pathway without or with little involvement of mitochondria and with caspase-8 activation and modulation of extrinsic apoptotic regulators, which influence sensitivity to nsEP [13]. To the best of our knowledge, few studies were based on high-throughput transcriptome sequencing in this gene regulatory mechanism issue after IRE in glioma. The transcriptome includes the complete set of transcripts in cells. Based on the generation of a high-throughput sequencing technology platform, transcriptome sequencing is used to comprehensively and quickly obtain almost all transcriptional sequence information of cells during a certain state. Unbiased high-throughput transcriptome sequencing seems to be the most valid approach to avoid any bias during pathways pre-selection.

Electric field strength threshold values for U251 cell apoptosis/necrosis were identified under stepwise increased field intensity. Cells of apoptosis induction by IRE and sham intervention were sampled (0–6 h) for high-throughput transcriptome sequencing. Furthermore, the obtained results disclosed mixed RCD forms simultaneously (including autophagy, necroptosis, and ICD) [14]. Here, we aimed to characterize the mechanisms of IRE-mediated RCD in vitro using the U251 cell line by high-throughput transcriptome sequencing. We then confirmed the sequencing results by qPCR and Western blot. We hypothesized that IRE-mediated RCD occurs via the AP-1–Bim-related signaling pathway. Mechanistic characterization of IRE-induced RCD will be important for translation into brain tumor clinical trials, in which IRE-mediated RCD will be exploited to reduce recurrence and inhibit the progression of GBM.

## 2. Materials and Methods

### 2.1. Cell Culture and IRE

U251 human malignant glioma cells were maintained in Dulbecco’s modified eagle medium (DMEM) (hg)-10% FBS in 5% CO_2_, 37 °C. Cells were seeded into 6-well plates before electroporation. Tumor cells were exposed to a flat electric field generated by a custom-made IRE pulse generator (HUIWEI MEDICAL Inc., Suzhou, China) after reaching 70–80% confluency, with field intensity incrementally increased. The pulse parameters were set as pulse duration 100 µs, frequency 1 Hz, 60 pulses, and unipolar. For DAPI staining, samples were incubated in 10% DAPI for 20 min before the intervention. Flat non-uniform/uniform electric fields were generated by two parallel-needle electrodes (blunt-ended, diameter: 0.4 mm, center-to-center distance: 1 cm, perpendicularly oriented to the plate bottoms, made with stainless-steel) or two parallel-plate electrodes (thickness: 0.4 mm, width: 2 cm, height: 4 cm, edge-to-edge distance: 1 cm, both vertically erected on the plate bottoms, made with stainless-steel) in 3 mL PBS at the center of every well of the 6-well plate. The pulse parameters were set as pulse duration 100 µs, frequency 1 Hz, 60 pulses, unipolar, and applied voltage of 1100 V on the needle electrodes or 200–730 V on the plate electrodes. Three-dimensional numerical simulations of the flat non-uniform/uniform electric fields were modeled using the finite element software (COMSOL Multiphysics 5.6, COMSOL Inc., Stockholm, Sweden, 2021).

### 2.2. Apoptosis and Necrosis Assay

Field strength threshold values for cell apoptosis/necrosis were identified by DNA fluorescence staining (YO-PRO-1/PI co-staining) and morphological changes 24 h post electroporation. YO-PRO-1 staining permits real-time cytofluorometric analysis of apoptosis with cell viability unaffected [15]. For YO-PRO-1/PI co-staining, samples were incubated in 5% YO-PRO-1/PI immediately following the electroporation. YO-PRO-1 and PI resulted in green and red staining of cell apoptosis and necrosis separately. Pictures were taken using a Nikon DS-Ri2 Zoom fluorescent microscope which can splice approximately 100 single fields of a normal microscope. Each picture showed the full view of a well in a 6-well cell culture cluster. Fluorescence intensity after flat uniform electric field treatment was analyzed using ImageJ software (ImageJ 1.46, the National Institutes of Health, Bethesda, MD, USA, https://imagej.nih.gov/ij/ (accessed on 1 July 2021)). S-shaped curves were obtained after all data were fitted by a nonlinear sigmoid curve regression using GraphPad Prism Software (Prism 7.00, GraphPad Software Inc., CA, USA, 2021, the x-axis displays the field intensity, and the y-axis shows the percentages of the fluorescent area in relation to the entire uniform electrical field-treated area) (https://www.graphpad.com/scientific-software/prism/ (accessed on 2 July 2021)). After electroporation, cells were re-cultured for another 24 h to further characterize their states by light microscopy.

### 2.3. RNA Extraction and Transcriptome Sequencing

Flat uniform electric fields of 520 V/cm were generated by the two parallel-plate electrodes in 3 mL PBS at the center of every well of the 6-well plate. Normal culture medium was added after IRE. Sham treatment was similarly handled in parallel with no current delivery. Samples were collected at 0 h after sham treatment, 0 h, 3 h, and 6 h after the electroporation. Cells exposed to the electric field were scraped off plates by a cell scraper and were harvested by centrifugation at 1000 rpm for 5 min. A total of 1 mL of TRIzol reagent was then added and mixed thoroughly by vortexing. Three replicates were taken for samples of each time point. All samples were stored at −80° until total RNA extraction. Total RNA was extracted using the mirVana™ miRNA ISOlation Kit (Ambion-1561) according to the instructions of the kit. The purity and quantification of RNA were assessed using the NanoDrop 2000 spectrophotometer (Thermo Scientific, Waltham, MA, USA). RNA integrity was evaluated using the Agilent 2100 Bioanalyzer (Agilent Technologies, Santa Clara, CA, USA). Libraries were constructed using the TruSeq Stranded mRNA kit (Illumina, San Diego, CA, USA) according to the manufacturer’s instructions. These libraries were sequenced on the Illumina transcriptome sequencing platform (NovaSeq 6000 Sequencing System).

### 2.4. Western Blot

Western Blot was performed using standard Western blot methods. c-Fos and Bim were detected using an Anti-c-Fos antibody (ab190289, Abcam, Cambridge, UK) and an Anti-Bim antibody (ab32158, Abcam). c-Fos and Bim primary antibodies were followed by a secondary horseradish peroxidase (HRP)-conjugated goat anti-rabbit IgG (H+L) antibody (A0208, Beyotime, Shanghai, China). β-actin was detected using beta Actin Antibody (HRP Conjugated) (AB2001, Abways Technology, Shanghai, China). Western blot data were analyzed using ImageJ software.

### 2.5. Quantitative PCR

The qPCR for validating the mRNA expression of *JUN*, *FOS,* and *Bim* in U251 (human malignant glioma cells) and LN229 cells (glioblastoma) was performed. In samples collected at 0 h after sham treatment, 0 h, 3 h, and 6 h after the electroporation, total RNA was extracted using a FastPure^®^ Cell/Tissue Total RNA Isolation Kit (RC101, Vazyme Biotech Co., Ltd., Nanjing, Jiangsu, China) according to the manufacturer’s instructions. First Strand cDNA was synthesized according to the instructions of the Thermo Scientific RevertAid First Strand cDNA Synthesis Kit (Thermo Scientific, USA). mRNA was quantified using the Bestar^®^ SybrGreen qPCR mastermix kit (DBI^®^ Bioscience, Ludwigshafen, German) according to the instruction manuals. The mRNA expression levels in the control groups were set at 1.00.

### 2.6. Statistics

In the present analysis, 12 samples (4 time points × 3 biological replicates) were collected for transcriptome sequencing. Raw reads were quality filtered using Trimmomatic to remove low-quality reads and obtain the clean reads (https://github.com/timflutre/trimmomatic (accessed on 20 May 2021)). Approximately 6.77~7.29 G of clean data for each sample were retained for subsequent analyses. The Q30 base distribution ranged from 94.57 to 94.83%. The average GC composition was 49.68 %. Reads were mapped to the human genome (GRCh38) using the HISAT2 (https://daehwankimlab.github.io/hisat2/ (accessed on 22 May 2021)). The alignment rate across the 12 samples to the reference genome ranged from 97.02 to 98.55%. On the basis of the alignments, the FPKM of each gene was obtained using Cufflinks [16], and read counts of each gene were acquired by HTSeqcount (https://htseq.readthedocs.io/en/master/ (accessed on 25 May 2021)). The *q*-value represents the corrected *p*-value using the Benjamini and Hochberg method [17]. The threshold for significantly differential expression was set at *q*-value < 0.05 and foldchange > 2 or foldchange < 0.5. Differential expression gene (DEGs) analysis was carried out using the DESeq (2012) R package (https://www.huber.embl.de/users/anders/DESeq/ (accessed on 25 May 2021)). Hierarchical cluster analysis of DEGs was performed to display the expression pattern of genes in different samples. GO enrichment and KEGG pathway enrichment analysis of DEGs were carried out, respectively, using R on the basis of the hypergeometric distribution. The possible RCD-related genes were screened by comparing DEGs to RCD gene lists from KEGG (Apoptosis: hsa04210, hsa04215; Autophagy: hsa04136, hsa04140; Necroptosis: hsa04217; Ferroptosis: hsa04216; Pyroptosis: not available). The potential immune-related genes were screened by comparing DEGs to gene lists from secondary classification terms (Immune disease and Immune system) based on KEGG. Human transcription factor names and their targets were obtained through hTFtarget (http://bioinfo.life.hust.edu.cn/hTFtarget#!/ (accessed on 5 October 2021)). The possible RCD-related transcription factors were screened by comparing the RCD-related genes from the present results with transcription factors from hTFtarget. Corresponding potential target genes were predicted by comparing transcription factor targets from hTFtarget with DEGs. An additional condition of searching the target genes and filtering pathways was set at gene expression levels (FPKM) ≥ 3 to avoid false-positive patterns. The STRING database (http://string-db.org (accessed on 9 October 2021)) was utilized to construct the PPI network of RCD-related DEGs, transcription factors, and corresponding target genes, with a combined score ≥ 0.4. The PPI data were demonstrated through Cytoscape (https://cytoscape.org/ (accessed on 9 October 2021)). 

## 3. Results

### 3.1. Electric Field Intensity Threshold for Apoptosis/Necrosis of U251 Glioma Cell Line

With the field intensity incrementally increased, the percentages of the green/red fluorescent area in relation to the entire uniform electrical field-treated area sharply increased from basal values to near-constant values (Figure 1A). Fine S-shaped curves were obtained after the nonlinear sigmoid curve regression fitting. Apoptosis and necrosis were induced with a 50% response after electroporation treatment (Field intensity ED50 value) of 299.5 ± 1.7 V/cm and 542.6 ± 1.6 V/cm (mean ± Std. Error), respectively. The threshold field intensity (initial cell response) of electroporation-induced apoptosis and necrosis was approximately 270 V/cm and 520 V/cm, respectively. Representative fluorescence pictures of unaffected (240 V/cm) (blue fluorescence), apoptotic (520 V/cm) (green fluorescence), and necrotic cells (640 V/cm) (red fluorescence) are provided (Figure 1B). Incubation continued for a further 24 h after electroporation treatment. Corresponding light microscopy pictures of the three different states are demonstrated (Figure 1C). Field intensity below the threshold field intensity of apoptosis induction did not exhibit significant cytotoxicity on the U251 cell line. The unaffected cells showed no apparent changes in cell density and cellular morphology. After being exposed to the field intensity between the threshold field intensity of apoptosis induction and necrosis induction, the apoptotic cells exhibited drastically different behaviors. These behaviors included membrane blebs formation, separation from the surrounding cells, weak extracellular attachments to the plates, and disintegration of the cell into apoptotic bodies. Field intensity beyond the threshold field intensity of necrosis induction leads to the coagulative necrosis of cells. A similar phenomenon was also observed after the cells were exposed to the flat non-uniform electric fields generated by two parallel-needle electrodes (Figure 1D). Green fluorescence shows that the cells underwent apoptosis, and similar apoptotic morphologies appeared after a further 24 h incubation. The cells stained with red fluorescence appeared to have coagulative necrosis. Other areas show that the cells were intact, as observed 24 h later using light microscopy. The experimentally observed distribution shapes and boundaries of the three states are similar to numerical simulations (solid red and yellow lines). However, the simulation-predicted threshold field intensity of electroporation-induced apoptosis and necrosis seemed to be approximately 500 V/cm and 800 V/cm, respectively.

### 3.2. Comprehensive Regulatory Transcription Changes of IRE-Induced Apoptosis

Cells of apoptosis induction by IRE (520 V/cm) and sham intervention were sampled (0–6 h) for high-throughput transcriptome sequencing. There are 33 differential expression genes at 0 h after electroporation vs. 0 h after sham treatment (14 upregulated and 19 downregulated), 612 differential expression genes at 3 h (505 upregulated and 107 downregulated), and 1500 differential expression genes at 6 h (1223 upregulated and 277 downregulated). Volcano plots were employed to depict the differences in gene expression (Figure 2A). The results of the hierarchical cluster analysis are displayed in the form of a heat map (Figure 2B).

The up- and downregulated differential expression genes were annotated into 64 groups based on the GO level2 entries classifications (Figure 3A). GO annotations consist of three categories of differential expression genes: biological process, molecular function, and cellular component. The top 10 GO terms, sorted by the -log10P-value and containing at least three differential expression genes, in the three categories were screened out, respectively (Figure 3B). GO term “transcription factor *AP-1* complex” (black rectangle) of “cellular component” is top-ranked. Additionally, the GO term “inflammatory response” (black rectangle) of “biological process” is also in the ranking list. Bubble plots are used to portray the top 20 enriched KEGG pathways, sorted by the -log10P-value and containing at least three differential expression genes (Figure 4). Inflammatory-related signaling pathways (IL-17 signaling pathway, MAPK signaling pathway, TNF signaling pathway, and NF-kappa B signaling pathway) and apoptosis-related signaling pathways (TNF signaling pathway and FoxO signaling pathway) are top-ranked.

### 3.3. Transcription Factors FOS and JUN Played a Major Role in the Apoptosis after IRE, Accompanied by Mixed RCD Forms

In this experiment, a total of 31 possible RCD-related genes were screened out by comparing DEGs to RCD gene lists from KEGG (not counting ICD-related genes). Distributions of the 31 genes among different RCD forms, including apoptosis, autophagy, necroptosis, and ferroptosis, are shown in Table 1. However, there are no readily available pyroptosis pathway maps from KEGG. The role that pyroptosis plays in the RCD after IRE remains unclear. No differential expression genes in the present study were involved in the ferroptosis pathway.

Among the 31 possible RCD-related genes, the screening identified two transcription factors: *FOS* and *JUN*. PPI data of the union of the 31 RCD-related genes, two transcription factors, and corresponding transcription factors target genes, were demonstrated (Figure 5). The foldchange of *JUN* was 1.43 at 0 h after electroporation (mean expression level 115) vs. 0 h after sham treatment (mean expression level 78) and did not meet the predetermined threshold for significantly differential expression (foldchange > 2 or foldchange < 0.5). This question aside, *FOS*, in conjunction with *JUN*, played a central role in the RCD, including apoptosis at all three time points after IRE.

A total of 122 potential immune-related genes were screened by comparing DEGs and gene lists from secondary classification terms (immune disease and immune system) based on KEGG. The temporal profile of the immune-related differentially expressed genes of all three time points after IRE is shown in Table 2.

Finally, we confirmed the sequencing results by qPCR and Western blot (Figure 6). The mRNA levels of *FOS* and *JUN*, major constituents of the *AP-1* transcription factor, and *Bim*, a known executioner of RCD, were all gradually upregulated after the treatment (Figure 6C–H). Expression of the protein c-Fos was also consistently upregulated (Figure 6A,B). These all echoed the sequencing results. However, the expression of the protein Bim was complicated (Figure 6A,B). Bim showed high basal expression and did not reduce instantly after the treatment. The protein amount of Bim then dropped to the lowest expression at 3 h and gradually upregulated 6 h later. This will be discussed in the ‘Discussion’ section. Overall, the qPCR and Western blot findings supported the sequencing results.

## 4. Discussion

The primary purpose of the present study was to comprehensively explore the mechanism of apoptosis after IRE based on high-throughput transcriptome sequencing. To confirm apoptosis, YO-PRO-1/PI co-staining and cellular morphological changes after re-culture for another 24 h were used (Figure 1B–D). YO-PRO-1 staining confirmed apoptosis in near real-time [18]. Cellular changes of typical apoptotic morphologies [19] 24 h later, including membrane blebs formation, separation from the surrounding cells, weak extracellular attachments to the plates, and disintegration of the cell into apoptotic bodies, further validated apoptosis.

However, the fact that COMSOL simulation-predicted threshold field intensity of electroporation-induced apoptosis and necrosis was approximately 500 V/cm and 800 V/cm, respectively, needs to be discussed in depth. We noticed that the cathode electrode of the two parallel-needle electrodes generated a large number of gas bubbles (Appendix A), while a similar phenomenon was not apparent on the two parallel-plate electrodes. The gas bubbles formation may arise from the excessive field strength generated around the needle electrodes. Most of the cathode electrode surface of the needle electrode was covered by the bubbles, causing local current density to increase, leading to local field intensity increasing, resulting in the experimental field strengths being smaller than the predicted values in other regions. The difference between the simulation-predicted apoptosis and necrosis threshold field intensity of electroporation (approximately 300 V/cm) is similar to the difference between the field intensity ED50 value for apoptosis and necrosis induction (243.1 V/cm). The overpredicted field strengths are likely to arise from gas bubble formation on the needle electrode. Field intensity ED50 values for apoptosis and necrosis induction obtained by utilizing the two parallel-plate electrodes are more robust. Like our results, electrode design for IRE should adequately consider the influence of bubble formation [20].

Through the optimized field intensity gradient settings of flat uniform electric fields, fine S-shaped curves were obtained (Figure 1A), which is similar to “dose-effect relationships” in pharmacological investigations. Apoptosis and necrosis sharply increased in a field intensity-dependent manner if the corresponding thresholds were exceeded. Originally, ED 50 was defined as the median effective dose, the dose of a drug that is effective in 50% of individuals [21]. Field intensity resembles the dose of the drug, and individuals refer to single cells here. For the first time, we draw on the pharmacological concepts of “S-shaped curves”, “dose-effect relationships”, and “ED50” to study the biological effects of electroporation.

The number of differentially expressed genes from samples after apoptosis induction by IRE increased progressively with time from 0 h to 6 h in a regulatory way. Profiles of transcriptome sequencing were unbiasedly demonstrated (Figure 2A,B, Figure 3A,B and Figure 4). The “transcription factor *AP-1* complex”, immune-related go terms, and signaling pathways were more prominent. Firstly, activator protein-1 (*AP-1*) is a classical regulator of cellular apoptosis in specific cell types [22]. *AP-1* consists of heterodimers and homodimers of *JUN*, *FOS*, or activating transcription factor bZIP proteins. However, to our knowledge, this is the first report of such a transcription factor in the gene regulatory mechanism of apoptosis after IRE. Secondly, what deserves attention is that apoptosis is usually immunologically silent [23], contrary to our transcriptome sequencing results, that 122 potential immune-related genes were involved. In-depth data mining for other forms of RCD [14] will be performed later. 

Possible differentially expressed RCD-related genes were screened out (Table 1). Although iron-based stainless-steel electrodes might generate in situ Fe^2+^ during electroporation, which plays an important role in ferroptosis [24], no differential expression genes were detected to be involved in the ferroptosis pathway in the present study. Involvement of ferroptosis in RCD within 6 h after IRE can therefore be ruled out. The possible RCD-related genes are distributed in different RCD forms, including apoptosis, autophagy, and necroptosis. Necroptosis is a highly immunogenic type of RCD [25], and autophagy has been reported to control further inflammatory responses [26]. Apoptosis is linked to DNA damage via several pathways that were not yet fully studied in the case of nsEP. One possibility is via PLK-1 protein and centrosome-mediated apoptosis, PUMA and NOXA were not activated. Exposure of cells to nsEP causes ER stress that could be related to ROS formation, permeabilization of the ER, or Ca^2+^ influx and can further trigger mitochondria-mediated intrinsic apoptosis via PERK and IRE1. ROS formation that occurs after nsEP could trigger both intrinsic and extrinsic apoptosis via several cellular targets, including those in the mitochondria, DNA, ER, and plasma membrane [13]. Among the 122 potential immune-related genes, 28 genes act as pro-inflammatory factors, and 4 genes act as anti-inflammatory factors [27]. The 28 pro-inflammatory genes we detected are 14 differentially expressed pro-inflammatory cytokines, cytokine receptors, and cytokine-induced proteins: *IL17B*, *IL17C*, *IL1A*, *IL23A*, *IL6*, *IL5RA*, *TNF*, *TNFRSF11A*, *TNFRSF13C*, *CCL2*, *CCL26*, *CXCL2*, *CXCL3,* and TNF alpha-induced protein 3 (*TNFAIP3*); pro-inflammatory complement receptors *C3AR1* and *C5AR1*; heat shock proteins *HSPA1A* and *HSPA1B*; Toll-like receptors *TLR5* and *TLR9*; leukocyte differentiation antigen: *CD14*, *CD1A*, *CD1D*, *CD74*, and *CD79A*; kinases *JAK3* and *MAPK13*; nuclear factor *NFATC2*. On the opposite side, four anti-inflammatory factors, including cytokine receptors (*TNFRSF10A* and *IL1R2*) and NFKB inhibitors (*NFKBIE* and *NFKBIA*), were detected. Although mainly produced by immune cells, inflammation-related substances produced by tumor cells are also common [28]. More promisingly, these inflammation-related substances after IRE endow the U251 glioma cells with the potential of generating a systemic antitumor immune response. The immunogenicity merits further thorough inquiry. As amply discussed above, the apoptosis induction of IRE caused multiple RCD forms (involvement of apoptosis, autophagy, necroptosis, and ICD transcriptome mechanisms) simultaneously.

Except for the fact that the foldchange of *JUN* at 0 h after the electroporation (mean expression level at 0 h after sham treatment as benchmark) did not meet the predetermined threshold for significantly differential expression, upregulation of *FOS* and *JUN*, major constituents of the *AP*-ranscription factor [22], maintained throughout all the three time points (0 h, 3 h, and 6 h). *AP-1* transcription factor was also the only transcription factor among the 31 possible RCD-related genes. GO term “transcription factor *AP-1* complex” of “cellular component” was top-ranked after GO enrichment analysis of all DEGs (Figure 3B). Electrical stimulation was reported to induce the upregulation of *AP-1* [29]. Additionally, increased *AP-1* activity can cause apoptosis in specific cell types, including tumor cells and neuronal cells [22]. Similarly, transcription factor *AP-1* should play a key role during apoptosis after the irreversible electroporation in this experiment. However, *FOS* expression duration and intensity in our study were considerably higher than the moderate one in the published data [29]. Su et al. reported a transient *FOS* expression elevation 1 h after electro-stimulation (approximate expression level = 10, fold change = 2), returning to baseline after 4 h, causing no significant cell death and excitotoxicity. Our results show an immediately early upregulation after the irreversible electroporation, a drastic upregulation in less than 3 h, and maintenance of high expression level for at least 6 h. It is widely believed that transcription factor *AP-1* mediates apoptosis by mainly regulating the gene expression of *P53* [30,31], *Fas* [32], *Fas-L* [32], *Bim* [33], and *HRK* [34]. Combining our results, *Bim*, also named Bcl2l11, was the only DEG with gene expression levels (FPKM) ≥ 3, whose expression level gradually increased and significantly upregulated at 3 h and 6 h. Putcha et al. [35] reported that the protein amount of *Bim* started to gradually upregulate 6 h after a cellular disturbance. The trend toward the upregulation of *Bim* from 3 h to 6 h (Figure 6A,B) after electroporation was consistent with the findings of Putcha et al. Importantly and different from them, the U251 cells showed high basal expression of *Bim* at 0 h after sham treatment (C group) and 0 h after electroporation (G0H group) (Figure 6A,B). We conjectured that the lowest expression of *Bim* at 3 h was due to protein leakage of basal *Bim* through cellular membrane holes formed after the electroporation. The upregulation of *Bim* around 6 h reflected de novo synthesis of protein *Bim*. However, more additional follow-up experiments need to be performed in the future.

In summary, our time-course transcriptome sequencing data suggest that IRE possibly mediate apoptosis by long-term high-intensity upregulation of transcription factor *AP-1* and upregulation of *Bim* (Bcl2l11) expression. Based on this prediction, further studies will need to be carried out on the pathways of IRE-induced apoptosis.

## Figures and Tables

**Figure 1 brainsci-12-01465-f001:**
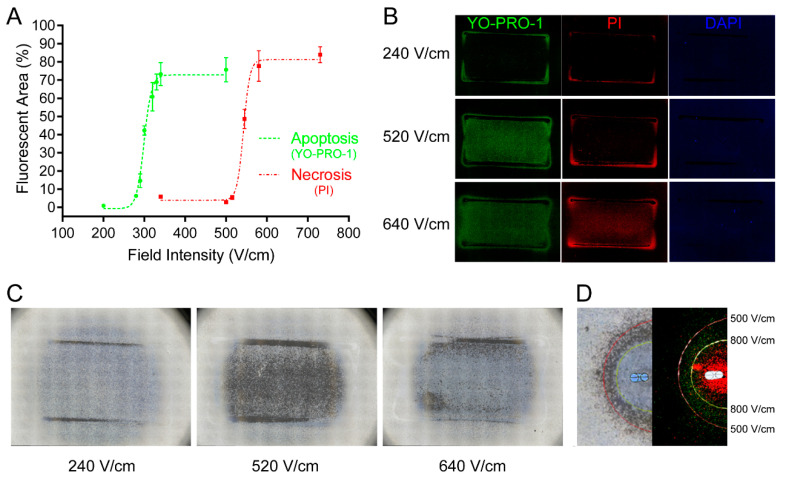
Electroporation-induced apoptosis and necrosis were determined by field intensity. (**A**) Field intensity of electroporation influenced the percentages of green (apoptosis) and red (necrosis) fluorescent areas to the entire treated area in a dose–effect-related manner. The threshold field intensity of electroporation-induced apoptosis and necrosis was approximately 270 V/cm and 520 V/cm, respectively. (**B**) Representative fluorescence images of unaffected (green and red fluorescence double negative), apoptotic (green fluorescence single positive), and necrotic cells (green and red fluorescence double positive) are presented. Nuclei of all cells were visualized using DAPI staining (blue fluorescence). (**C**) Corresponding representative light microscopy pictures of unaffected (240 V/cm), apoptotic (520 V/cm), and necrotic cells (640 V/cm) were acquired 24 h after electroporation. (**D**) Similar fluorescence after electroporation (right part) and corresponding morphological changes 24 h afterwards (Left part) were reproduced by the flat non-uniform electric fields generated by two parallel-needle electrodes. Solid red and yellow lines are numerical simulations of rough apoptotic and necrotic threshold field intensity (500 V/cm for the red line and 800 V/cm for the yellow line). Perspective blue and white cylinders represent one of the parallel-needle electrodes.

**Figure 2 brainsci-12-01465-f002:**
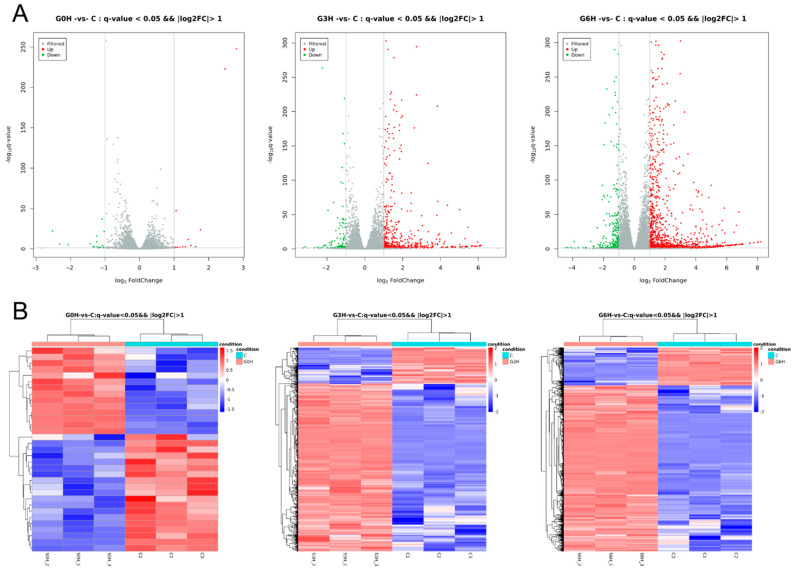
Time series transcriptomic changes of electroporation-induced apoptosis revealed gradually widespread gene expression differences. (**A**) Volcano plots for showing time series differentially expressed genes (*q*-value < 0.05 and foldchange > 2 or foldchange < 0.5). (**B**) Heat maps for displaying hierarchical cluster analysis of time series differentially expressed genes.

**Figure 3 brainsci-12-01465-f003:**
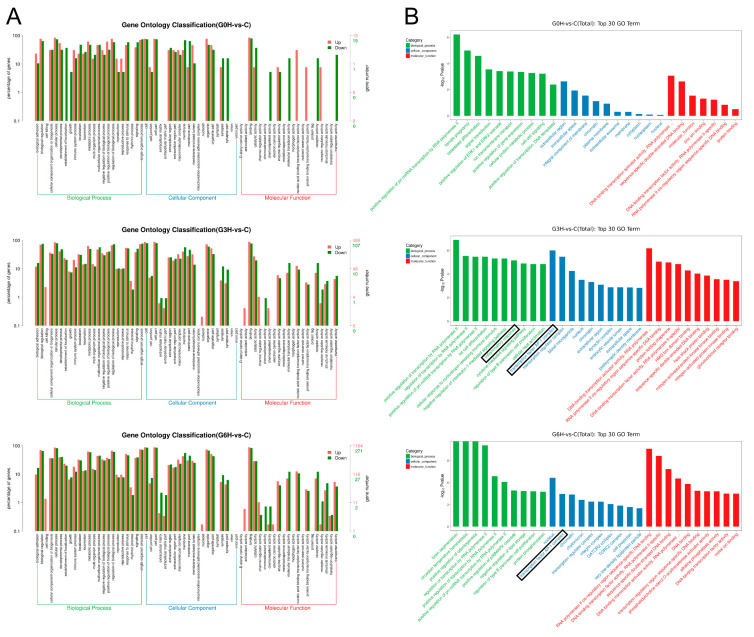
Data mining of the time series transcriptomic changes of electroporation-induced apoptosis. (**A**) Gene ontology (GO) term analysis was performed to annotate the up- and downregulated differential expression genes into 64 groups based on the GO level2 entries classifications. (**B**) GO terms were sorted by the -log10P-value, and the top 10 GO terms in each of the three GO level1 categories were displayed. The term "transcription factor *AP-1* complex” of “cellular component" and “inflammatory response” of “biological process” was depicted by a black rectangle.

**Figure 4 brainsci-12-01465-f004:**
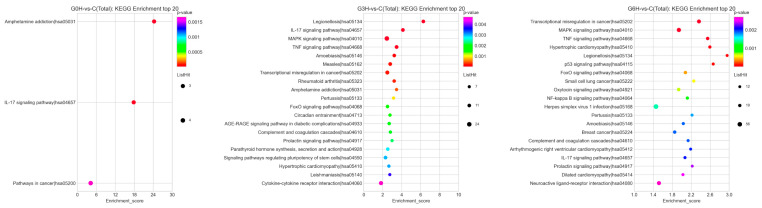
Enriched KEGG pathways were sorted by the -log10P-value, and the top 20 pathways were portrayed using bubble plots.

**Figure 5 brainsci-12-01465-f005:**
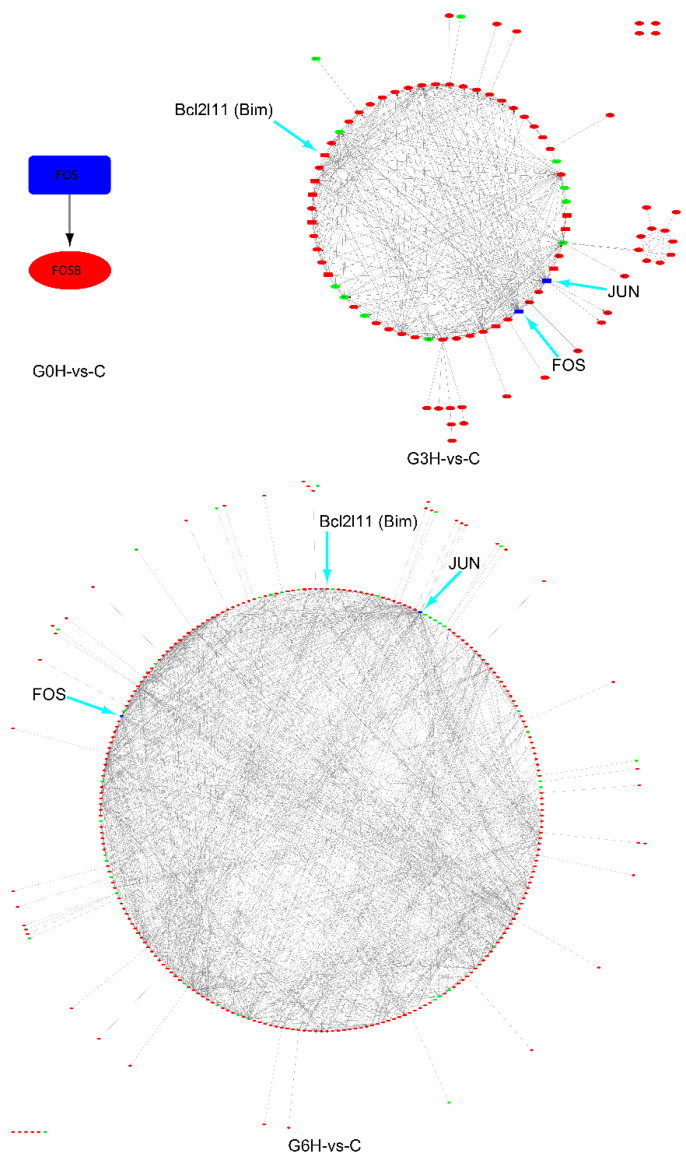
Time series differential expression genes interactions network of the 31 RCD-related genes (not counting ICD-related genes), 2 transcription factors, and corresponding target genes of the transcription factors. Blue nodes represent upregulated transcription factors; red nodes are upregulated genes; green nodes are downregulated genes; round rectangle nodes are RCD-related genes; ellipse nodes are non-RCD-related genes; edges are genes interactions; edge directions are the regulatory interactions from transcription factors to their target genes. Cyan arrowheads indicate amplified gene names on the gene interaction networks.

**Figure 6 brainsci-12-01465-f006:**
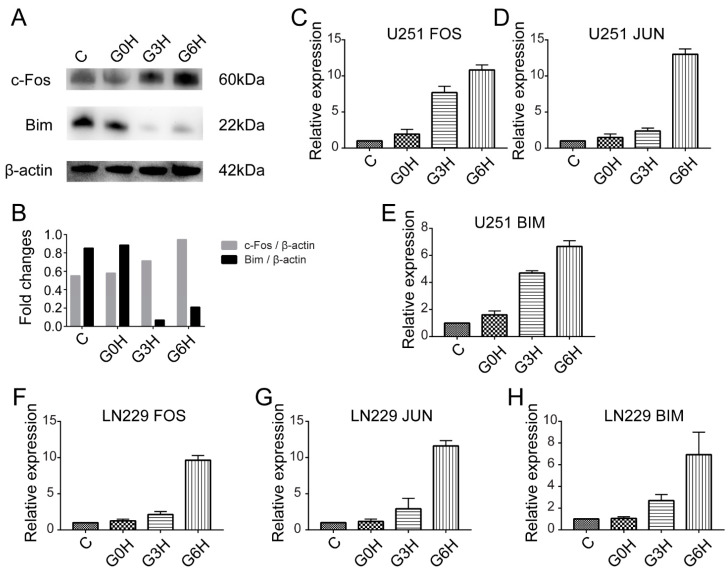
Upregulation of transcription factor *AP-1* and upregulation of *Bim* (Bcl2l11) expression after electroporation. Western blot results (**A**) and analysis (**B**) of *c-Fos* and *Bim* in samples collected at 0 h after sham treatment, 0 h, and 6 h after electroporation in U251 cells. *c-Fos* was consistently upregulated. *Bim* showed high basal expression and did not significantly reduce instantly after the treatment and then dropped to the lowest expression 3 h later. Expression of *Bim* was gradually upregulated 6 h later. qPCR results (**C**–**H**) of *FOS* (**C**,**F**), *JUN* (**D**,**G**), and *Bim* (**E**,**H**) in two different cells, U251 (**C**–**E**) and LN229 cells (**F**–**H**), all showed gradual upregulation after electroporation, which was consistent with the sequencing results.

**Table 1 brainsci-12-01465-t001:** Distributions of the screening 31 possible RCD-related differential expression genes among RCD forms (not counting ICD-related genes).

Apoptosis	Apoptosis-Multiple Species	Autophagy-Other	Autophagy-Animal	Necroptosis	Pyroptosis	Ferroptosis
hsa04210	hsa04215	hsa04136	hsa04140	hsa04217	Not Available	hsa04216
BIRC3	BIRC3			BIRC3	N/A	/
BCL2L11 APAF1	BCL2L11 APAF1					
PIK3R3 EIF2AK3			PIK3R3 EIF2AK3			
TNFRSF10A TNF				TNFRSF10A TNF		
		ATG9B	ATG9B			
			DDIT4 DEPTOR			
GADD45B MCL1 JUN DDIT3 PIDD1 GADD45A NFKBIA FOS NTRK1 GADD45G ATF4						
				H2AC6 PLA2G4C H2AC20 PYGM JAK3 PLA2G4F IL1A USP21 TNFAIP3 ZBP1		

KEGG pathway names and entries were listed at the head of the table. “N/A” represented that there were no currently available pyroptosis pathway maps from KEGG. “/” represented that there was no distribution of RCD-related DEGs in "Ferroptosis" in this experiment.

**Table 2 brainsci-12-01465-t002:** Time course transcriptomic changes of electroporation-induced RCD to involve immune-related differentially expressed genes.

G0H-vs-C	G3H-vs-C	G6H-vs-C
**FOS** FOSB	**FOS** FOSB	**FOS** FOSB
MMP1	MMP1	
	FERMT3 NFKBIA ITGAM MAPK13 H2BU1 ATP6V1B1 GRIN2B C5AR1 HSPA6 **EGR3** SERPINC1 TNFAIP3 H2BC18 CCL2 IL17B TNFRSF10A CXCL3 C3AR1 DLL1 MYLK3 IL6 PRKCG MAP3K8 PTPN6 IL23A PECAM1 CXCL2 **EGR2** **BCL3** IL1A ITGAX H3C4 IL17C C9 **JUN** CD1D CD79A BUB1B-PAK6 IL1R2 GNB3 RASGRP2 H2BC17 JAK3 HSPA1B PLA2G4F RASSF5 CD14 HSPA1A	FERMT3 NFKBIA ITGAM MAPK13 H2BU1 ATP6V1B1 GRIN2B C5AR1 HSPA6 **EGR3** SERPINC1 TNFAIP3 H2BC18 CCL2 IL17B TNFRSF10A CXCL3 C3AR1 DLL1 MYLK3 IL6 PRKCG MAP3K8 PTPN6 IL23A PECAM1 CXCL2 **EGR2** **BCL3** IL1A ITGAX H3C4 IL17C C9 **JUN** CD1D CD79A BUB1B-PAK6 IL1R2 GNB3 RASGRP2 H2BC17 JAK3 HSPA1B PLA2G4F RASSF5 CD14 HSPA1A
CCL26		
	HSPA2 FGG COL3A1 F3 CD74 APBB1IP	
		KIT TNFRSF13C H2BC6 H2BC5 H2BC4 CD34 C2 **TBX21** DLL3 H3C13 SCIN CXCR4 MASP2 GRK7 GBP1 PLA2G4C **FOSL1** KSR1 H2AC20 MYLK4 F2 **FOXO3** H2BC11 POLR1C TNFRSF11A **RUNX1** CD1A PTGS2 BDKRB2 CALML4 TNF FLT1 LCK TLR9 H2AC6 **RARA** DLL4 GUCY1A2 NFATC2 CLDN1 TBKBP1 TLR5 IL5RA F2RL2 POLR2H RAG1 **IRF1** F2R CLDN19 NFKBIE **IRF3** SERPINF2 PIK3R3 BIRC3 JAM2 THBD ITGB7 H3C3 ITGA2B H2BC8 NAMPT ZBP1 PIP5K1A BTK

Head of the table shows the sampling time points. Red font denotes transcription factors.

## Data Availability

Data can be available upon request by contacting the corresponding author via email.

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
