# Peer review of "Irreversible Electroporation Mediates Glioma Apoptosis via Upregulation of AP-1 and Bim: Transcriptome Evidence"

_brainsci, 2022, doi:10.3390/brainsci12111465_

Round 1
Reviewer 1 Report (Previous Reviewer 4)
As suggested, the authors have made improvements to the manucript notably in the introduction and the discussion but I have still some concerns especially for the results section.
Major points
Contrary to what the authors say, the PCRq and western blot figures presented in their response to the reviewers are not present in the revised version of the manuscript. It the second time that a figure is missing in the manuscript, the author should be more careful. And I think that these figures need to be described and commented in the results section which is not the case here. Notably, BIM is not up regulated according to the quantification of the western blot which doesn't match their hypothesis.
I think that the microscope used and the method for imaging should be added in the materials and methods section for better comprehension of the images presented in figure 1.
Minor points
The resolution of Figure 3 is better, but it is still low and the text is still blurry after magnification.
Author Response
Major points
- Contrary to what the authors say, the PCRq and western blot figures presented in their response to the reviewers are not present in the revised version of the manuscript. It the second time that a figure is missing in the manuscript, the author should be more careful. And I think that these figures need to be described and commented in the results section which is not the case here.
Answer: Thank you for your suggestion. In the revised manuscript, we have added the PCRq and western blot figures into the manuscript as figure 6.
- Notably, BIM is not up regulated according to the quantification of the western blot which doesn't match their hypothesis.
Answer:Thank you for your suggestion. We have discussed this point in depth in the ‘Discussion’ section in the revised manuscript.
- I think that the microscope used and the method for imaging should be added in the materials and methods section for better comprehension of the images presented in figure 1.
Answer:Thank you for your suggestion. We have added the microscope used and the method for imaging in detail in the materials and methods section in the revised manuscript.
Minor points
- The resolution of Figure 3 is better, but it is still low and the text is still blurry after magnification
Answer:Thank you for your suggestion. In the revised manuscript, we have revised the Figure 3 with a higher resolution.
Reviewer 2 Report (Previous Reviewer 2)
A good revision of the paper was done.
Author Response
Answer: Thank you for your evaluation.
Round 2
Reviewer 1 Report (Previous Reviewer 4)
As suggested, the authors have made improvements to the manuscript that seem sufficient for publication in brain science.
This manuscript is a resubmission of an earlier submission. The following is a list of the peer review reports and author responses from that submission.
Round 1
Reviewer 1 Report
The paper described properly
Author Response
Thank you for your evaluation.
Reviewer 2 Report
This topic sounds very interesting,but some points need to be revised:
- Lines 70-74: It is not clear what is the aim of this paper. Please improve this point.
- Lines 37-38: "GBM patients only have a median overall survival of 12-15 months[2] and inevitably die of tumor recurrence or progression" , and despite a second surgical treatment. Consider these 2 very important and recent refs: -- doi: 10.1007/s13760-021-01765-4 -- doi: 10.1093/neuonc/nov326
- Lines 250-257: "Among the 31 possible RCD-related genes, the screening... time points after IRE." This from a clinical point of view, what does it imply?
- Lines 317-319: "Secondly, what deserves attention is that apoptosis.. immune-related genes were involved. " what does it mean that it remains silent? please explain it better.
- Lines 369-375: "In summary, our time-course transcriptome sequencing... pathway of IRE-induced apoptosis in detail". When the reader reaches the end of the paper he/she does not yet understood, in my opinion, whether this paper is a review or a clinical / laboratory study.
- Lines 288-289: "Considering the fact that the difference between the simulation-pre... " Why do authors report these data at these point? What do they want to highlight?
- Lines 211-212: Figure 2 "(B) Heat maps for displaying hierarchical cluster analysis of time series differentially expressed genes." Please explain better these differentially expressed genes.
Author Response
Reviewer 2:
This topic sounds very interesting, but some points need to be revised:
- Lines 70-74: It is not clear what the aim of this paper is. Please improve this point.
Answer: Thank you for your suggestion. In the revised manuscript, we have add the aim of this paper in the introduction (Lines 70-74).
- Lines 37-38: "GBM patients only have a median overall survival of 12-15 months[2] and inevitably die of tumor recurrence or progression" , and despite a second surgical treatment. Considering these 2 very important and recent refs: -- doi: 10.1007/s13760-021-01765-4. Acta Neurol Belg. 2022 Apr;122(2):441-446 -- doi: 10.1093/neuonc/nov326. Neuro Oncol. 2016 Apr;18(4):549-56. We have added the two references in the revision.
- Lines 250-257: "Among the 31 possible RCD-related genes, the screening... time points after IRE." This from a clinical point of view, what does it imply?
Answer: From a clinical point of view, IRE could induce FOS/JUN associated target gene expression and mediated RCD will be exploited to reducing recurrence and inhibiting progression of GBM.
- Lines 317-319: "Secondly, what deserves attention is that apoptosis.. immune-related genes were involved. " what does it mean that it remains silent? please explain it better.
Answer:
According to the reference 21, presence of apoptotic cells increases their secretion of the anti-inflammatory and immunoregulatory cytokine interleukin 10 (IL-10) and decreases secretion of the pro-inflammatory cytokines tumour necrosis factor-a(TNF-a), IL-1 and IL-12. Thus, apoptosis may impaired cell-mediated immunity in conditions such as viral infections, pregnancy, cancer and exposure to radiation.
- Lines 369-375: "In summary, our time-course transcriptome sequencing... pathway of IRE-induced apoptosis in detail". When the reader reaches the end of the paper he/she does not yet understood, in my opinion, whether this paper is a review or a clinical / laboratory study.
Answer: Thank you for your suggestion. In the reviesd manuscript, we have revised this part of conclusion. We also added some validation experiment of western blot and qPCR.
- Lines 288-289: "Considering the fact that the difference between the simulation-pre... " Why do authors report these data at these point? What do they want to highlight?
Answer: We wanted to highlight that cathodal electrode of the two parallel needle electrodes generated a large number of gas bubbles. The gas bubbles causing local current density increasing, leading to local field intensity increasing, resulting in that the experimental field strengths were smaller than the predicted values in other regions. In the future clinical usage, we should reduce and consider the production and the influence of gas bubbles.
- Lines 211-212: Figure 2 "(B) Heat maps for displaying hierarchical cluster analysis of time series differentially expressed genes." Please explain better these differentially expressed genes.
Answer: Heat maps for displaying hierarchical cluster analysis of time series differentially expressed genes. The differentially expressed genes including 33 differential expression genes at 0h after the electroporation (14 up-regulated and 19 down-regulated), 612 differential expression genes at 3h (505 up-regulated and 107 down-regulated), and 1500 differential expression genes at 6h (1223 up-regulated and 277 down-regulated).
Reviewer 3 Report
Yu et al studied the genetic mechanisms and forms of regulatory cell death in a glioma cell line that was exposed to the irreversible electroporation. Their findings indicated that AP1-Bim pathway has roles in IRE-mediated apoptosis. Their findings may advance our understanding of surgical ablation methods and improve methods towards minimal side-effects, including post-operative neurological deficits. The manuscript may benefit from the comments below:
Major issues:
1) The entire study relies on U251 cell line. Most, if not all, journals now require data from at least 3 different cell lines. Therefore, the significance and accuracy of the findings presented in this manuscript is questionable.
2) Extending from the previous comment, several studies now showed that the cell lines that are commonly used by many laboratories worldwide turned out to be contaminated by several other commonly-used cell lines. Have the authors performed a cell line identification method?
3) The Discussion section tend to have very long sentences (e.g., lines 273-276, lines 292-296). These sentences and others can be divided into two or even three.
Minor issues:
1) Please divide the material and methods section into subheadings, as it is relatively challenging to read 1.5 pages of plain text.
2) In my opinion, the manuscript may benefit from increasing the figure sizes (e.g. Fig. 4).
Author Response
Reviewer 3:
Major issues:
1) The entire study relies on U251 cell line. Most, if not all, journals now require data from at least 3 different cell lines. Therefore, the significance and accuracy of the findings presented in this manuscript is questionable.
Answer:Considering the high cost of transcript sequencing(12 samples (4 time points, we selected one cell line U251(3 biological replicateswere collected for accuracy) as model. In the revision, We used another other two cell lines for verification by qPCR.
2) Extending from the previous comment, several studies now showed that the cell lines that are commonly used by many laboratories worldwide turned out to be
contaminated by several other commonly-used cell lines. Have the authors performed a cell line identification method?
Answer:Thank you for your suggestion. The U251 cell line we used was newly bought from Sigma-Aldrich (Human: U251MG 09063001). Another paper using the same cell line close in time was provided: https://doi.org/10.1016/j.cmet.2021.03.003. Cell Metab 2021 05 04;33(5).
3) The Discussion section tend to have very long sentences (e.g., lines 273-276, lines 292-296). These sentences and others can be divided into two or even three.
Answer: Thank you for your suggestion. In the revised manuscript, we have adjusted the long sentences in the discussion.
Minor issues:
1) Please divide the material and methods section into subheadings, as it is relatively challenging to read 1.5 pages of plain text.
Answer:Thank you for your suggestion. In the reviesd manuscript, we have adjusted the material and methods section.
2) In my opinion, the manuscript may benefit from increasing the figure sizes (e.g. Fig. 4).
Answer:Thank you for your suggestion. In the reviesd manuscript, we have increasing the figure resolution in figure 4.
Reviewer 4 Report
The authors analyzed by high-throughput transcriptome sequencing in the malignant glioma cell line U251, the genes induced by IRE, a soft tissue ablation technique using electric fields, to determine whether it is a suitable therapy for GBM. They showed that apoptosis and necrosis were induced after the application of a field intensity gradient with a "dose-response" effect. Then, they chose the dose of 520V/cm to study differentially expressed genes after IRE-induced apoptosis after 0-6 h. The article describes the results obtained.
This is a rather preliminary work that requires further experiments to add value to the results obtained.
General comments
All abbreviations should be defined when first used in the text (L.40: 52, 76).
In vitro should be written in italics.
Figures 1 and 3 are too small. Figure 3: it is impossible to read the text on the pdf. When enlarging it to the maximum, it becomes blurred.
Figure 5 is missing: it is not in the text and is not attached separately.
L.359 to 368 I think this part should be in the result section. What are the numbers for bim? What does "significantly up-regulated" mean?
Specific comments:
1) The introduction is rather short maybe a paragraph on the signaling pathways regulating cell death after IRE in others cancers would be a plus.
2)Figure 1B: aren't we supposed to see the nuclei with Dapi? All I see is a uniform blue background.
Figure 1C: I don't understand what we are supposed to see here: where are the intact cells?
3) A least, simple PCRq experiment confirming the transcriptomic results obtained for JUN, FOS, some of the pro-inflammatory genes and bim is crucial. In addition, a western blot showing upregulation of these proteins would add weight to the described results.
4) Discussion should be improved: comparaison with pathways found by others in the literature, interest of the identified signaling pathways
